# A Novel Salt Inducible Kinase 2 Inhibitor, ARN-3261, Sensitizes Ovarian Cancer Cell Lines and Xenografts to Carboplatin

**DOI:** 10.3390/cancers13030446

**Published:** 2021-01-25

**Authors:** Dengxuan Fan, Hailing Yang, Weiqun Mao, Philip J. Rask, Lan Pang, Congjian Xu, Hariprasad Vankayalapat, Ahmed A. Ahmed, Robert C. Bast, Zhen Lu

**Affiliations:** 1Department of Experimental Therapeutics, University of Texas M.D. Anderson Cancer Center, Houston, TX 77054, USA; dxfan16@fudan.edu.cn (D.F.); HYang3@mdanderson.org (H.Y.); WMao@mdanderson.org (W.M.); PRask54775@med.lecom.edu (P.J.R.); LPang@mdanderson.org (L.P.); 2Obstetrics and Gynecology Hospital, Fudan University, Shanghai 200011, China; xucongjian@fudan.edu.cn; 3Department of Obstetrics and Gynecology of Shanghai Medical College, Fudan University, Shanghai 200011, China; 4Shanghai Key Laboratory of Female Reproductive Endocrine Related Diseases, Fudan University, Shanghai 200011, China; 5Arrien Pharmaceuticals, 9980 South, 300 West, Suite # 200, Salt Lake City, UT 84070, USA; hari@hci.utah.edu; 6The Nuffield Department of Obstetrics and Gynecology, University of Oxford, Oxford OX3 9DU, UK; ahmed.ahmed@wrh.ox.ac.uk; 7Oxford NIHR Biomedical Research Centre, Oxford OX4 2PG, UK

**Keywords:** salt inducible kinase 2 (SIK2), PARP inhibitor, carboplatin sensitivity, γ-H2AX, comet assay, apoptosis

## Abstract

**Simple Summary:**

Carboplatin and paclitaxel constitute first-line treatment for ovarian cancer, producing tumor shrinkage in 70% of patients, but curing less than 20% with advanced stage disease. Previous studies have shown that treatment with ARN-3261, a small molecule inhibitor of the enzyme salt-induced kinase 2, can improve the response to paclitaxel in human ovarian cancer cells grown in culture and in immunocompromised mice. Here we have found that ARN-3261 also increases carboplatin’s ability to kill ovarian cancer cells grown in culture and in immunocompromised mice, causing additional DNA damage and decreasing levels of survivin, a protein that protects cancer cells from programmed cell death. Our studies encourage clinical evaluation of ARN-3261 and a Phase I clinical trial has been initiated to test the drug in ovarian cancer patients.

**Abstract:**

Salt-induced kinase 2 (SIK2) is a serine-threonine kinase that regulates centrosome splitting, activation of PI3 kinase and phosphorylation of class IIa HDACs, affecting gene expression. Previously, we found that inhibition of SIK2 enhanced sensitivity of ovarian cancer cells to paclitaxel. Carboplatin and paclitaxel constitute first-line therapy for most patients with ovarian carcinoma, producing a 70% clinical response rate, but curing <20% of patients with advanced disease. We have asked whether inhibition of SIK2 with ARN-3261 enhances sensitivity to carboplatin in ovarian cancer cell lines and xenograft models. ARN-3261-induced DNA damage and apoptosis were measured with γ-H2AX accumulation, comet assays, and annexin V. ARN-3261 inhibited growth of eight ovarian cancer cell lines at an IC50 of 0.8 to 3.5 µM. ARN-3261 significantly enhanced sensitivity to carboplatin in seven of eight ovarian cancer cell lines and a carboplatin-resistant cell line tested. Furthermore, ARN-3261 in combination with carboplatin produced greater inhibition of tumor growth than carboplatin alone in SKOv3 and OVCAR8 ovarian cancer xenograft models. ARN-3261 enhanced DNA damage and apoptosis by downregulating expression of survivin. Thus, a SIK2 kinase inhibitor enhanced carboplatin-induced therapy in preclinical models of ovarian cancer and deserves further evaluation in clinical trials.

## 1. Introduction

Ovarian cancer is a leading cause of gynecological cancer death. Each year, 230,000 women will be diagnosed with ovarian cancer and 150,000 will die from the disease worldwide [1]. High-grade serous ovarian cancer (HGSOC) accounts for 70–80% of ovarian cancer deaths, and long-term survival has not changed significantly for several decades [2]. Most patients are treated with cytoreductive surgery and combination chemotherapy using carboplatin and paclitaxel. Seventy percent of patients with primary disease experience a clinical response, but <20% of patients can be cured with advanced stage disease.

Our group has sought kinases that regulate the response of ovarian cancer cells to paclitaxel and carboplatin and whose inhibition might improve outcomes for women with ovarian cancer [3]. One of the most promising targets to date is salt-induced kinase 2 (SIK2) which is overexpressed in 30% of ovarian cancers, associated with decreased progression-free survival [4]. SIK2 belongs to the AMPK family [5]. It is a serine-threonine kinase that regulates centrosome splitting [4], facilitates cell-cycle progression [6], actives PI3 kinase [7,8,9], reprograms glucose and fatty acid metabolism [10,11], and phosphorylates class IIa HDACs [12], thus affecting gene expression.

Arrien Pharmaceuticals has developed novel 1H-(pyrazol-4-yl)-1H-pyrrolo [2,3-b] pyridine inhibitors, ARN-3236 and ARN-3261, which compete for ATP binding to SIK2 protein and inhibit SIK2 kinase activity. ARN-3236 inhibits SIK2 activity with an IC50 <1 nM, but does not significantly inhibit the other two SIK family members, SIK1 and SIK3, as well as other AMPK family members [13]. ARN-3236 is, however, susceptible to efflux by the P-glycoprotein (P-gp) transporter. ARN-3261, a clinical lead compound derived from ARN-3236 by introducing a solvent binding sulfone, showed acceptable profiles in cell-based proliferation assays, ADME and in PK/PD studies and resisted efflux through the P-gp transporter. Thus, for clinical use, ARN-3261 appeared more promising than ARN-3236.

In our earlier studies, inhibition of SIK2 with ARN-3236 enhanced the sensitivity of ovarian cancer cells to paclitaxel in cell culture and in xenografts [14]. As the primary target of platinum drugs is DNA, sensitivity or resistance to treatment is affected by the ability of cells to recognize and repair drug-induced DNA damage. In this report we have asked whether ARN-3261 could increase DNA damage and enhance response to carboplatin.

## 2. Results

### 2.1. ARN-3261 Inhibits Cell Growth and Increases Sensitivity to Carboplatin in Ovarian Cancer Cells

To determine whether ARN-3261 could inhibit the growth of ovarian cancer cells, the effect of ARN-3261 was measured in eight ovarian cancer cell lines using short term cell proliferation assays. The IC50 of ARN-3261 was calculated for each cell line (Figure 1A). Significant inhibition was achieved in all cell lines in a dose dependent manner. The IC50 values of ARN-3261 for OVCAR8, SKOv3, OC316, OVCAR3, ES2, A2780, MDA2774, and IGROV1 cells ranged from 0.8 to 3.5 µM. The IC50 values for carboplatin with the same ovarian cancer cell lines ranged from 1.2 to 34.2 µM (Figure 1B). When ARN-3261 was added to carboplatin, the carboplatin dose response curve was shifted to the left in seven of eight ovarian cancer cell lines (Figure 1C), indicating that ARN-3261 sensitizes ovarian cancer cells to carboplatin (*p* < 0.01). To test whether the interaction was additive or synergistic, we performed multi-point drug combination studies in four of the most responsive cell lines (IGROV1, OC316, OVCAR8 and SKOv3), and calculated a combination index (CI) using the Chou-Talaley method, based on a median-effect equation to define the drug response to the combination quantitatively. CI values in response to the ARN-3261 and carboplatin combination were less than one in all four cell lines (Figure 1D and Appendix A), supporting the hypothesis that SIK2 inhibition enhances sensitivity to carboplatin.

Furthermore, to exclude potential off-target effects of ARN-3261, we knocked down SIK2 with CRISPR/Cas9 in OVCAR8 and SKOv3 ovarian cancer cells. Knockout of SIK2 sensitized ovarian cancer cells to carboplatin in a manner similar to ARN-3261 (Figure 1E). In addition, five cell lines were used to compare the effect of ARN-3261 and carboplatin to either agent alone using clonogenic assays, which showed that ARN-3261 significantly enhanced carboplatin induced loss of clonogenic survival in the OVCAR8, SKOv3, OC316, MDA4772 and ES2 ovarian cancer cell lines (Figure 1F and Appendix A). Taken together, these data suggest that the inhibition of SIK2 kinase activity potentiates carboplatin in ovarian cancer cells, and ARN-3261, a potent SIK2 selective inhibitor, works synergistically with carboplatin to kill ovarian cancer cells.

### 2.2. ARN-3261 or SIK2 Knockout Enhances Carboplatin-Induced Apoptosis by Downregulating Survivin

Many current cancer chemotherapies, including platinum-based drugs, exert their antitumor effect by triggering apoptosis in cancer cells [15]. To study the underlying mechanism of SIK2 inhibition-induced carboplatin-mediated cell toxicity, apoptosis was measured using flow cytometry in OVCAR8, SKOv3 and OC316 ovarian cancer cell lines. ARN-3261 not only induced apoptosis as a single agent, but also enhanced carboplatin-induced apoptosis (Figure 2A). In addition, a similar effect was observed in SIK2 knockout cell lines (OVCAR8 and SKOv3) showing that abolishing the function of SIK2 enhanced ovarian cancer cells to carboplatin-mediated apoptosis (Figure 2B). Together, our data suggest that SIK2 inhibition enhances carboplatin sensitivity by increasing carboplatin-induced apoptotic cell death.

The inhibitor of apoptosis protein family (IAPs) includes an important group of proteins involved in the regulation of apoptosis [16]. One member of this protein family, survivin, plays an important role in promoting tumor progression by deregulating apoptosis and cell division [16]. In our study, downregulation of survivin was observed with ARN-3261 and greater downregulation was observed with the combination of the two drugs in OC316 and OVCAR8 ovarian cancer cell lines (Figure 3). When SIK2 was knocked out using CRISPR/cas9, cells expressed less survivin and the downregulation phenotype with ARN-3261 treatment was partly reversed (Figure 3, right panel). Thus, downregulation of survivin and a consequent activation of apoptosis could contribute to ARN-3261-mediated carboplatin sensitization.

### 2.3. ARN-3261 Enhances Carboplatin-Induced DNA Damage

The biochemical mechanism(s) for cytotoxicity of cisplatin and carboplatin involve covalent binding to DNA and induction of cell death through apoptosis within the heterogeneous population of tumor cells [17,18]. Direct binding of platinum-based drugs to genomic DNA in cancer cells can result in a number of lesions including bulky platinum-DNA adducts and DNA double-strand breaks (DSBs) [19]. Detection of increased γ-H2AX punctae is an early and sensitive indicator of DSBs after treatment with cisplatin or carboplatin in cancer cells [20,21]. As carboplatin induces apoptosis, we examined whether treatment with ARN-3261 increases carboplatin-induced γ-H2AX punctae in OVCAR8, SKOv3 and OC316 ovarian cancer cell lines. ARN-3261 and carboplatin showed a greater increase in γ-H2AX punctae than either agent alone (Figure 4A). In addition, a comet assay was also performed to measure DNA damage after treatment with ARN-3261, carboplatin or the combination of two drugs. Consistent with an increase in γ-H2AX punctae, the comet tail moment induced with carboplatin was further enhanced by ARN-3261 (Figure 4B). Thus, it suggests that ARN-3261 enhances carboplatin-mediated apoptosis by increasing carboplatin-induced DNA damage.

### 2.4. ARN-3261 Inhibits Growth Of Cisplatin-Resistant Cancer Cell Lines and Enhances Sensitivity to Carboplatin

Platinum resistance is commonly seen in ovarian cancer patients with recurrent disease. There is currently no standard chemotherapy for platinum-resistant recurrence. Targeting DNA damage and repair is an attractive therapeutic approach in platinum-resistant ovarian cancer [22]. As ARN-3261 enhances carboplatin-induced DNA damage, we tested whether SIK2 inhibition with ARN-3261 will overcome platinum-induced resistance in ovarian cancer cells. A2780-PAR cisplatin-sensitive and A2780-CP20 cisplatin-resistant ovarian cancer cells were tested for carboplatin response and the IC50 of A2780-CP20 (34.9 µM) was 32-fold higher than IC50 of A2780-PAR (1.1 µM) (Figure 5A). ARN-3261 inhibited both the resistant and sensitive cell lines in a dose dependent fashion with IC50’s of 2.4 and 0.6 µM, respectively (Figure 5B). Treatment with the combination provided synergistic enhancement of the carboplatin effect as CI values were <1 (Figure 5C,D). Thus, ARN-3261 enhanced sensitivity to carboplatin not only in carboplatin-sensitive ovarian cancer cells but also in carboplatin-resistant ovarian cancer cells.

### 2.5. ARN-3261 Enhances the Activity of Carboplatin in Human Ovarian Cancer Xenograft Models

Given the synergistic effect of ARN-3261 and carboplatin in inhibiting the growth of cultured ovarian cancer cells, we investigated whether the addition of the SIK2 inhibitor could promote carboplatin response in xenograft models. OVCAR8 cells were injected intraperitoneally (ip) into nu/nu mice. Treatment started 7 days post injection. ARN-3261 (50 mg/Kg) was administered orally five days a week while carboplatin (25 mg/kg) was injected ip once a week for three weeks. At the conclusion of the treatment, the mice were sacrificed, and the tumor was dissected and weighed (Figure 6A). Treatment with ARN-3261 significantly enhanced the growth inhibitory effect of carboplatin (*p* < 0.05) (Figure 6B). Moreover, the combination of ARN-3261 with carboplatin was well tolerated, with no significant weight loss compared to vehicle control (Figure 6B). To validate results observed in the OVCAR8 xenograft model, SKOv3 cells were injected subcutaneously into nu/nu mice. Seven days after tumor cell injection, mice were treated with either vehicle, single-agent ARN-3261 (40 mg/kg), carboplatin (10 mg/kg) or paclitaxel (50 mg/kg), or the combination of two or three drugs as indicated for a total of 6 weeks (Figure 6C). The tumor volume was measured at indicated time points (Figure 6D). Treatment with ARN-3261, carboplatin, or paclitaxel alone significantly inhibited tumor growth (*p* < 0.001) (Figure 6D), compared to vehicle control. The combination of ARN-3261 plus carboplatin (*p* < 0.05) or ARN-3261 plus paclitaxel (*p* < 0.01) produced greater inhibition of tumor growth than either single agent (Figure 6D). More importantly, ARN-3261 further enhanced the combination treatment of carboplatin plus paclitaxel (*p* < 0.01) (Figure 6D) which is standard first-line chemotherapy for patients with ovarian cancer.

## 3. Discussion

In this report, we have found that ARN3261 induces double strand breaks (DSBs) in cancer cell DNA and produces synthetic lethality with carboplatin. SIK2 is an AMP-activated protein kinase that is required for ovarian cancer cell proliferation and metastasis. SIK2 is overexpressed in 30% of ovarian cancers, correlating with poor prognosis in patients with high-grade serous ovarian carcinomas. ARN-3261 inhibited cancer cell growth in eight ovarian cancer cell lines with IC50 concentrations that ranged from 0.8 to 3.5 µM. ARN-3261 enhanced carboplatin sensitivity in seven of eight ovarian cancer cell lines and in two xenograft models. ARN-3261 significantly increased carboplatin-mediated γ-H2AX production and DNA comet tail moment, indicating enhanced DNA damage and/or decreased DNA repair. In addition, treatment with ARN-3261 sensitized both a relatively sensitive A2780-parental cell line and the highly resistant A2780-CP20 cell line, demonstrating that ARN-3261 enhanced sensitivity to carboplatin both in carboplatin-sensitive and in carboplatin-resistant ovarian cancer cells.

Platinum-based drugs including cisplatin, carboplatin, and oxaliplatin are widely used for the treatment of different cancers. Treatment with a combination of paclitaxel and carboplatin is considered first line therapy for advanced ovarian cancer. Ovarian cancer responds well to both cisplatin and carboplatin, but after an initial response, the majority of patients with ovarian cancer will relapse and develop the resistance. Because the main target of platinum drugs is DNA, the sensitivity and resistance to those drugs is associated with the ability of cells to repair the platinum-induced DNA damage [23,24]. ARN-3261 was found to enhance carboplatin-induced DNA damage judged by γ-H2AX accumulation and an increase in comet assay tail moment. Enhancement of DNA damage was associated with an increase in apoptosis that was most pronounced with the combination of ARN-3261 and carboplatin. This combination also downregulated survivin. Recent studies show that survivin is associated with both inhibiting apoptosis and regulating cell mitosis in cancer [14,25]. Survivin overexpression has been shown to correlate with chemo-resistance in several cancers [26]. Several molecular approaches that downregulate survivin expression and/or block its function are being developed in the clinic [16]. Our past and present findings indicate that the SIK2 inhibitor ARN-3261 enhances sensitivity to both carboplatin and paclitaxel in cultured ovarian cancer cell lines as well as in xenograft models, supporting its potential role in the treatment of primary as well as recurrent ovarian cancer.

Taken together, our studies encourage the further clinical evaluation of ARN-3261. A phase I study of ARN-3261 (GRN-300) alone and in combination with paclitaxel is underway. Pre-clinical toxicology studies indicate that treatment with ARN-3261 has little effect on normal hematopoietic or organ function. In the present study, treatment with ARN-3261 and carboplatin did not affect the body weight of nude mice. If ARN-3261 is well tolerated in the current phase I study, a phase I-II study could be initiated with ARN-3261 and carboplatin.

## 4. Materials and Methods

### 4.1. Reagents

ARN-3261 was provided by Arrien Pharmaceuticals (Patent No. US-9260426-B2). The purity is 98.2%. The drug was dissolved in DMSO at 10 mM as a stock for in vitro assays. The final concentration of DMSO was <1%. ARN-3261 was dissolved in 5% of ethanol, 30% of polyethylene glycol-300 and 2% of Tween 80 (*v*/*v*) by sonication for in vivo animal studies. Carboplatin and paclitaxel were purchased from MD Anderson Pharmacy at 10 mg/mL and 6 mg/mL, respectively. Carboplatin was prepared in sterile water and diluted in culture media for in vitro assays. For in vivo animal studies, drugs were diluted in sterile saline to desired concentrations.

### 4.2. Cell Lines and Cultures

OVCAR8, SKOv3, OC316, OVCAR3, ES2, A2780, IGROV1 and MDA2774 human ovarian cancer cell lines were provided by Dr. Gordon Mills’ laboratory (UT MD Anderson Cancer Center, Houston, TX, USA). SKOv3 WT and SKOv3-SIK2 KD (clone 1D) cell lines, OVCAR8 WT and OVCAR8-SIK2 KD (clone 2-3A) cell lines were provided by Dr. Ahmed A. Ahmed (Oxford University, Oxford, UK). A2780-PAR and A2780-CP20 were kindly provided by Dr. Sood at MD Anderson. The STR DNA fingerprinting was performed at MD Anderson (Characterized Cell line Core). In addition, mycoplasma was tested in the cell lines using Universal Mycoplasma Detection Kit (ATCC^®^ 30-1012K) and all cell lines were free from contamination. RPMI1640 was used for culturing OVCAR8, SKOv3, OC316, OVCAR5, OVCAR3, ES2, IGROV1, MDA2774, OVCAR8 WT and OVCAR8-SIK2 KD cells. McCoy’s 5A was used for culturing SKOv3 WT and SKOv3-SIK2 KD. Both RPMI1640 and McCoy’s 5A were purchased from the Media Preparation Core Facility at MD Anderson Cancer Center.

### 4.3. Cell Viability Assays

Cells were seeded in 6 replicates in black-walled and clear-bottomed 96-well plates and incubated overnight. Cells then were treated with ARN-3261 and/or carboplatin for an additional 4 days using the concentrations indicated in each figure. The CellTiter-Glo luminescent cell viability assay (Promega) was used to evaluate the effect of treatment on cancer growth [27,28]. We performed this experiment several times to optimize the concentration. To study the interaction of drugs, we reduced the ARN-3261 concentration incrementally starting from the concentration equal to the IC50 value of ARN-3261 used as a single agent in Figure 1A. And the concentrations shown above can shift the carboplatin dose-response curve to the left, indicating improved drug responses. GraphPad Prism 8 was used to generate growth curves and calculated IC_50_. CalcuSyn was used to evaluate additive or synergistic interactions, a combination Index (CI value) [29]. Values < 1 are considered synergisitic and Values >1 or =1 are additive or sub-additive.

### 4.4. Clonogenic Survival Assays

Cancer cells were seeded in 6-well plates at a density of 400 cells per well in culture medium for 24 h to permit cell adherence. Subsequently, cells were treated with ARN-3261 and/or carboplatin in triplicate. After treatment, cells were grown for an additional 12–14 days. After control colonies had grown to include at least 50 cells, cultures were fixed and stained with Coomassie blue (0.1% Coomassie brilliant blue R-250, 40% methanol, and 10% acetic acid) and counted. Colonies were counted from three independent experiments and the mean number of colonies and standard deviations calculated. Multiplicity adjusted *p* values for each treatment and control were determined.

### 4.5. Protein Extraction and Western Blot Analysis

Cells were incubated for 24–48 h with and without treatments and then harvested for western blot analysis. Briefly, cells were incubated in lysis buffer for 20 min on ice and centrifuged at 17,000× *g* for 10 min at 4 °C. Protein concentration of cell lysates was determined with BCA reagent (Thermo Fisher Scientific, Houston, TX, USA). Lysates were separated on 8–16% SDS-PAGE and transferred to polyvinylidene difluoride (PVDF) membranes. Immunoblots were probed with anti-survivin antibody (Novus; 1:2000, Centennial CO, USA) in 5% BSA overnight at 4 °C and HRP labeled secondary antibody was added for 1 h at RT. The signal was developed on X-ray films. The uncropped western blotting figures can be found in Appendix A.

### 4.6. Immunofluorescence Staining

Cells were seeded on coverslips in 12-well plates with or without treatment as indicated in each figure. Cells were then fixed in 4% paraformaldehyde (Affymetrix, Sunnyvale, CA, USA) for 10 min, permeabilized with 0.1% triton X-100 in PBS for 15 min, and then blocked with 1% BSA in PBS for 1 h at RT followed by incubation with anti-r-H2AX at 1:500 dilution (Cell Signaling) at 4 °C overnight. Coverslips were washed 3 times with PBS after primary rabbit antibody incubation and incubated with anti-rabbit Ig secondary antibody conjugated to Alexa 488 for 1 h (Life Technologies, A11070, 1:200, Austin, TX, USA). Cells were rinsed and the nuclei stained with DAPI (Thermo Fisher, 1 µg/mL). Cells were examined using fluorescence microscopy (Olympus 1 × 71; Olympus Corporation of the Americas, Center Valley, PA, USA).

### 4.7. Comet Assays

The OxiSelectTM Comet Assay Kit (Cell Biolabs, Inc., San Diego, CA, USA) was used to evaluate DNA damage with or without drug treatment. Briefly, cells (1 × 10^5^ cells/mL) were mixed with molten agarose (Cell Biolabs, Inc.) at 37 °C at a ratio of 1:10 (*v*/*v*), and then transferred to comet slides and incubated in the dark for 15 min. Slides were immersed in pre-chilled lysis buffer for 1 h and then with freshly prepared pre-chilled Alkaline Solution (pH > 13) for 30 min at 4 °C in the dark. Slides were electrophoresed in alkaline buffer at 1 volt/cm for 30 min. The cells were stained with 1x Vista Green DNA Dye for 15 min at RT and then viewed with a 4× fluorescence microscope. The percentage of DNA in the tail were analyzed with Casplab_1.2.3b2 (CaspLab Comet Assay Software, CASPLab, http://casplab.com/) [30]. At least 50 randomly selected cells were analyzed per sample.

### 4.8. Apoptosis Assays

Cells were grown in 6-well plates at a density of 8 × 10^4^ cells/plate and treated with or without ARN-3261 and/or carboplatin. After completion of incubation, cells were harvested and washed with PBS two times. After washing, cells were re-suspended in 100 μL Annexin-binding buffer containing propidium iodide (PI) and FITC Annexin-V (Invitrogen) and incubated at ambient temperature for 15 min in the dark. After incubation, 200 uL of Annexin-binding buffer was added and stained cells were analyzed using a Beckman Coulter’s Gallios Flow Cytometer.

### 4.9. Murine Xenografts

Six-week-old female athymic nu/nu mice were purchased from Envigo. Experiments were reviewed and all procedures were performed according to an animal protocol approved by the Institutional Animal Care and Use Committee of UT MD Anderson Cancer Center. OVCAR8 ovarian cancer cells (3 × 10^6^) were inoculated i.p. and SKOv3 ovarian cancer cells (5 × 10^6^) were inoculated s.c. For OVCAR8 xenograft models, ARN-3261 was administrated p.o. at a dose of 50 mg/kg/day for 5 days/week for three weeks. Carboplatin was administrated i.p. once a week at a dose of 25 mg/kg. On day 7 after cancer cell injection, mice were randomly assigned to the following treatment groups (n = 10 mice per group): (1) non-treatment diluent control; (2) ARN-3261; (3) carboplatin; and (4) a combination of carboplatin and ARN-3261. For SKOv3 xenograft models, ARN-3261 was treated with at a dose of 40 mg/kg/day for six weeks. Carboplatin was treated at a dose of 30 mg/kg once a week for six weeks. Additionally, paclitaxel was administrated i.p. once a week at a dose of 0.8 mg/kg for once a week for six weeks. One week after cells injection, ten mice per group were randomly assigned to the following six groups: (1) diluent control; (2) ARN-3261; (3) carboplatin; (4) paclitaxel; (5) a combination of carboplatin and ARN-3261; (6) a combination of paclitaxel and ARN-3261; (7) a combination of carboplatin and paclitaxel; and (8) a combination of ARN-3261, carboplatin and paclitaxel. At end of six weeks, the mice were sacrificed by CO_2_. The mice were dissected immediately after death and the tumors were collected and weighed.

### 4.10. Statistical Analysis

If not stated otherwise, all experiments were set up as triplicates and repeated independently at least twice and the data were expressed as the mean ± the standard deviation. GraphPad Prism (version 8.0) was used for plotting and statistical analyses. The two-tailed Student *t* test (2 groups with unequal variances) and one-way ANOVA for multiple comparisons were performed. The differences at *p* < 0.05 was considered statistically significant.

## 5. Conclusions

SIk2 inhibitor ARN-3261 enhances sensitivity to carboplatin of both carboplatin-sensitive and resistant ovarian cancer cells in vitro, inhibits tumor xenograft growth and enhances sensitivity to both carboplatin and paclitaxel in in vivo xenograft models.

## Figures and Tables

**Figure 1 cancers-13-00446-f001:**
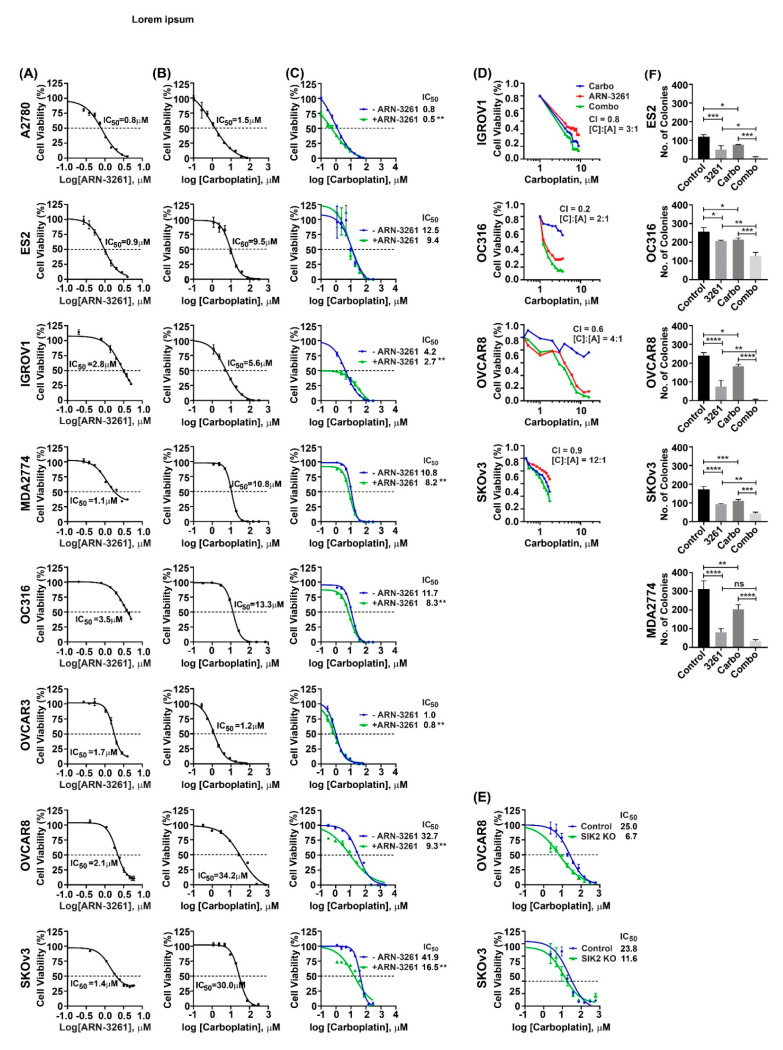
ARN-3261 synergistically enhances carboplatin-induced inhibition of ovarian cancer cell short term and clonogenic growth in cell culture. (**A**) Sensitivity to ARN-3261. A2780, ES2, IGROV1, MDA2774, OC316, OVCAR3, OVCAR8 and SKOv3 ovarian cancer cell lines were plated at a density of 2000 cells/well in 96-well plates, then treated with different concentrations of ARN-3261 for 96 h. Cell viability was measured with a bioluminescence assay as described in Methods and IC50 values were calculated. (**B**) Sensitivity to carboplatin. Eight ovarian cancer cell lines were treated as above with different concentrations of carboplatin as indicated. (**C**) Effect of a single concentration of ARN-3261 on the carboplatin dose response curve. Eight ovarian cancer cell lines were treated with different concentrations of carboplatin as indicated with or without a single concentration of ARN-3261 (A2780 0.75 µM, ES2 1.25 µM, IGROV1 1.25 µM, MD2774 1.15 µM, OC316 0.75 µM, OVCAR3 0.75 µM, OVCAR8 1 µM and SKOv3 1 µM). IC50s of carboplatin with or without ARN-3261 were calculated by GraphPad Prism 8 (** *p* < 0.01 by student *t* test). (**D**) Synergistic interaction of carboplatin and ARN-3261. IGROV1, OC316, OVCAR8 and SKOv3 were treated concomitantly with a serial dilution of ARN-3261 and carboplatin at a fixed ratio indicated in the figure. The drug concentration ratio is indicated in each plot. The combination index at 50% growth inhibition was calculated using CalcuSyn software. (**E**) Effect of SIK2 knockout on the carboplatin dose response curve. Cells were treated with different concentrations of carboplatin as indicated. IC50 values for (**A**–**C**,**E**) were calculated by GraphPad Prism 8. (**F**) ARN-3261 enhances carboplatin-induced inhibition of clonogenic growth. Four hundred OVCAR8 or SKOv3 ovarian cancer cells were seeded in 6-well plates in culture medium for 24 h. Cells were then treated with diluent, ARN-3261 (ES2 2.2 µM, OC316 2.5 µM, OVCAR8 2.3 µM, SKOv3 3.5 µM and MDA2774 2.5 µM), carboplatin (ES2 3.3 µM, OC316 3.0 µM, OVCAR8 4.0 µM, SKOv3 2.0 µM and MDA2774 3.0 µM) or both in triplicate for another 12–14 days. The graphs indicate the mean colony formation numbers with standard deviations. Statistical significance is indicated by * *p* < 0.05, ** *p* < 0.01, *** *p* < 0.001 and **** *p* < 0.0001 by one-way ANOVA analysis.

**Figure 2 cancers-13-00446-f002:**
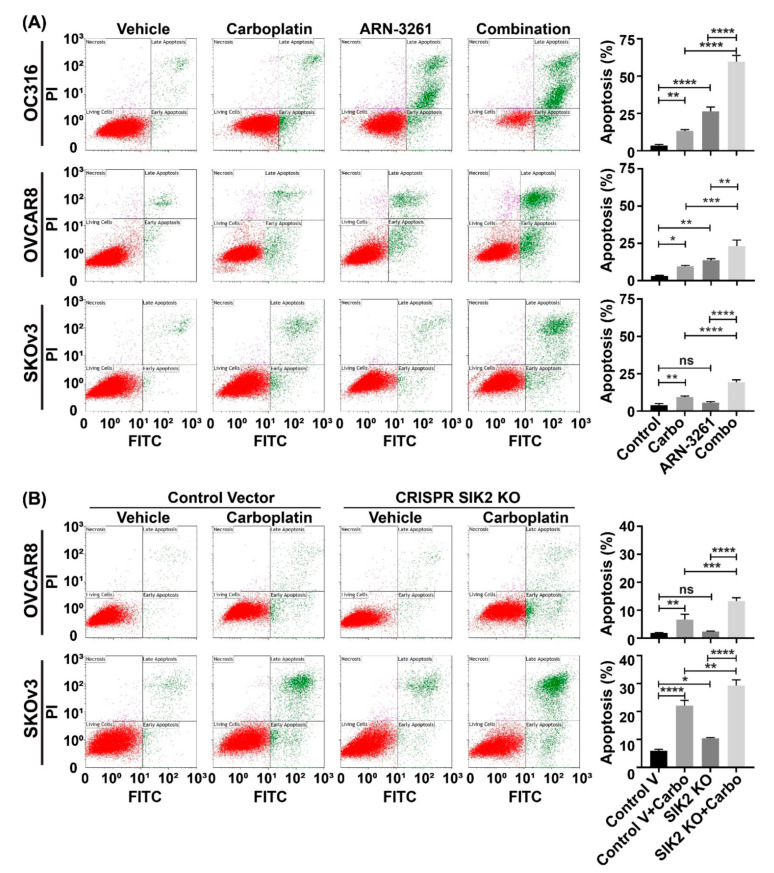
Inhibition of SIK2 activity with ARN-3261 or knockout of SIK2 protein enhances carboplatin-induced apoptosis. (**A**) Effect of ARN-3261 on carboplatin-induced apoptosis. OC316, OVCAR8 and SKOv3 cell lines were plated at a density of 8000 cells/well in 12-well plate in triplicate, and then treated with ARN-3261 (OC316 3 µM, OVCAR8 5 µM and SKOv3 4.5 µM) and/or carboplatin (OC316 15 µM, OVCAR8 70 µM and SKOv3 60 µM) for 72 h. Cells were dislodged and stained with Annexin V antibody and PI dye for flow cytometry. Representative images are shown on the left and the analysis of apoptotic population under different treatment conditions are on the right. (**B**) Effect of SIK2 knockout on carboplatin-induced apoptosis. SIK2 knockout (KO) and control cell lines were treated as in (**A**) and analyzed for apoptosis. The bars indicate the mean percentage of apoptotic cells with standard deviations. Statistical significance is indicated by * *p* < 0.05, ** *p* < 0.01, *** *p* < 0.001 and **** *p* < 0.0001. ns: not significant by one-way ANOVA analysis.

**Figure 3 cancers-13-00446-f003:**
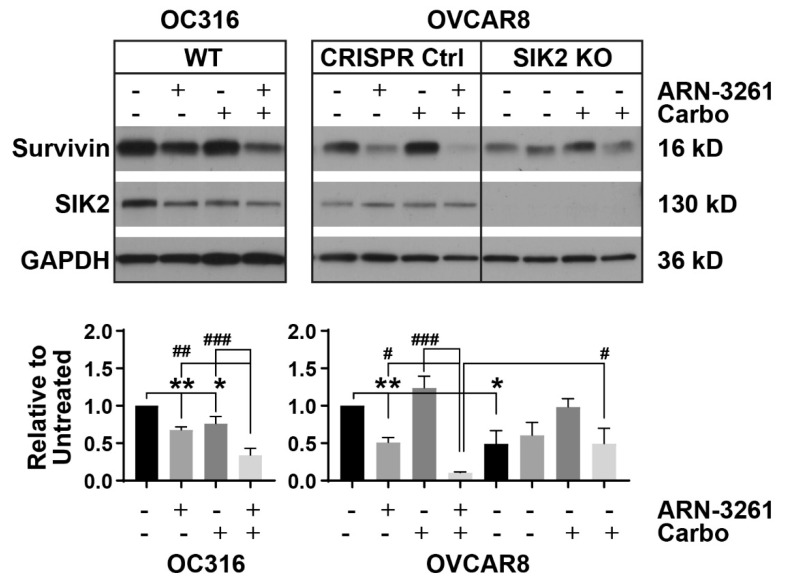
Treatment with ARN-3261 enhances the carboplatin-induced decrease in survivin expression. OC316, OVCAR8, and OVCAR8 SIK2 KO ovarian cancer cells were treated with diluent, ARN-3261 (OC316 3 µM and OVCAR8 5.0 µM), carboplatin (OC316 15 µM and OVCAR8 60 µM) and the combination for 48 h. Cell lysates were collected and survivin expression was measured by western blot analysis. The experiments were performed three times individually. Densitometry values were determined by Image J shareware (NIH) and normalized to the GAPDH loading control. The values relative to the untreated group were plotted at the bottom. Different treatments were compared by one-way ANOVA analysis. * *p* < 0.05 and ** *p* < 0.01 compared to untreated control group; # *p* < 0.05, ## *p* < 0.01, and ### *p* < 0.001 compared to the combination treatment of ARN-3261 and carboplatin.

**Figure 4 cancers-13-00446-f004:**
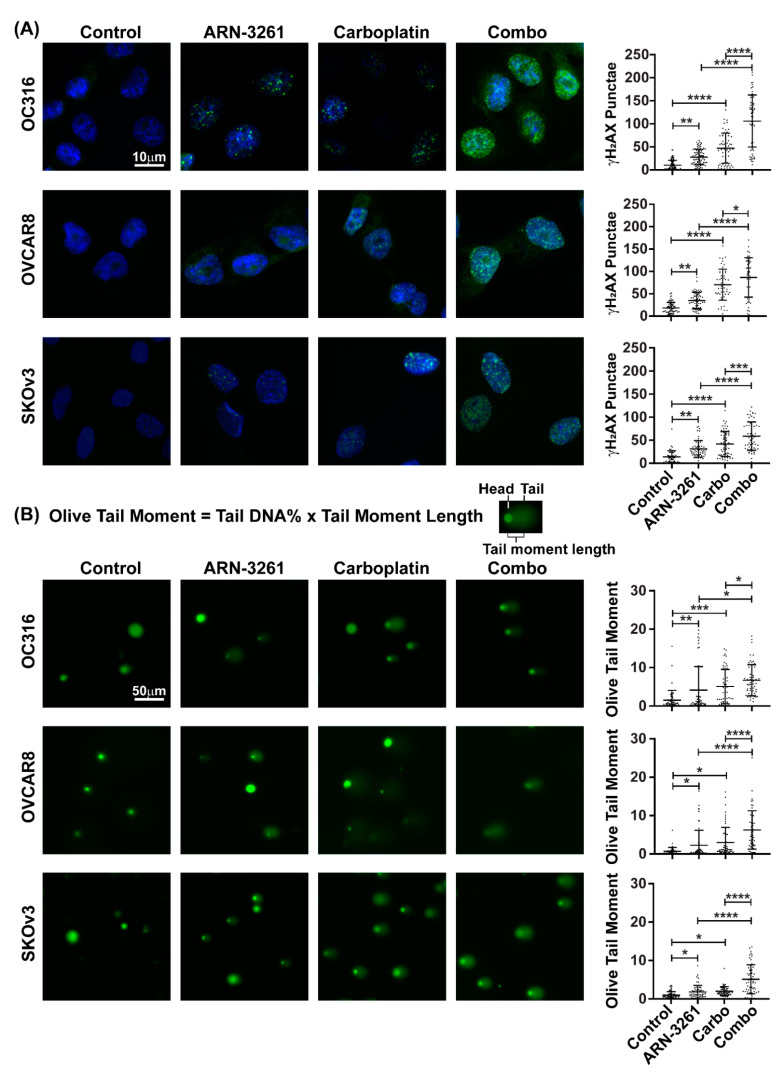
ARN-3261 enhances carboplatin-induced DNA damage. (**A**) OC316, OVCAR8 and SKOv3 ovarian cancer cells were treated with diluent, ARN-3261 (OC316 3 µM, OVCAR8 3.0 µM and SKOv3 3.5 µM), carboplatin (OC316 15 µM, OVCAR8 35 µM and SKOv3 35 µM) or the combination for 8 h and stained for γ-H2AX in green and for DNA with DAPI in blue. Each plot depicts the mean number of punctae (the bars indicate the standard deviation). (**B**) Cells were treated as described in (**A**) for 24 h. Then cells were dislodged, immobilized in agarose gel onto glass slide, and lysed. DNA was eletrophoresed in alkaline buffer and stained by Vista Green. Olive tail moment (OTL) was measured as described in the Methods. Each plot depicts the mean of OTL (the bars indicate the standard deviation). Statistical significance by one-way ANOVA is indicated by * *p* < 0.05, ** *p* < 0.01, *** *p* < 0.001 and **** *p* < 0.0001.

**Figure 5 cancers-13-00446-f005:**
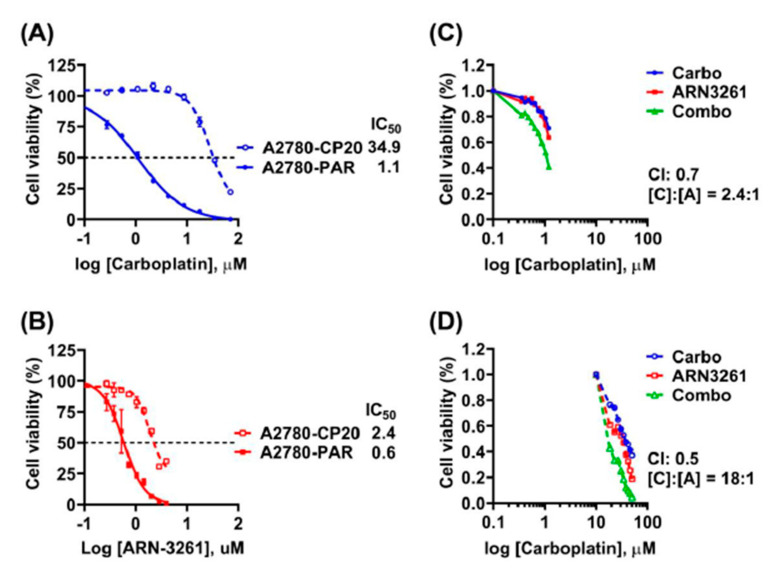
Treatment with ARN-3261 enhances carboplatin toxicity in cisplatin-sensitive and cisplatin-resistant sublines. (**A**) A cisplatin-resistant ovarian cancer cell subline is also resistant to carboplatin. Cisplatin-resistant A2780-CP20 and cisplatin-sensitive A27801 PAR sublines were plated at a density of 2000/well in 96-well plates, then treated with different concentrations of carboplatin for 96 h as indicated. Cell viability was measured with a bioluminescence assay and the IC50 was calculated. (**B**) Growth of both cisplatin-resistant and cisplatin-sensitive sublines are inhibited by ARN-3261 as indicated. Cells were similarly cultured and treated for 96 h with different concentrations of ARN-3261, before measuring cell viability and calculating IC50. (**C**,**D**). The interaction of ARN-3261 and carboplatin in cisplatin sensitive (**C**) and cisplatin resistant cell lines was evaluated by the combination index at 50% growth inhibition using CalcuSyn software.

**Figure 6 cancers-13-00446-f006:**
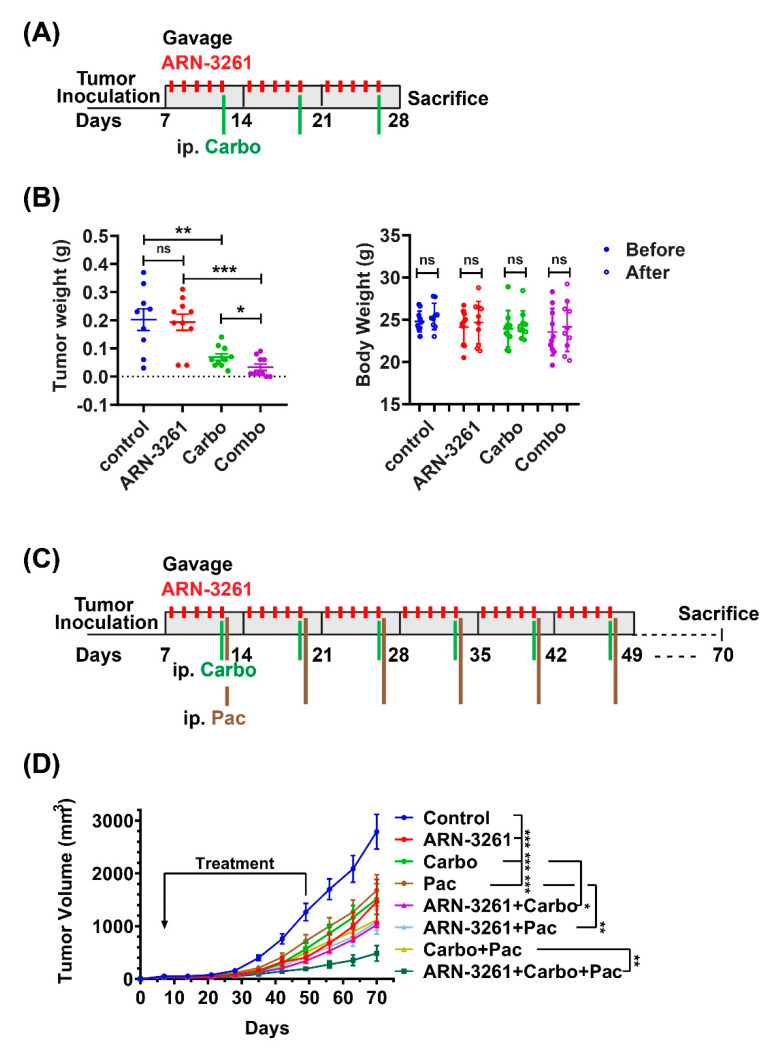
ARN-3261 enhances the activity of carboplatin in human ovarian cancer cell xenografts. (**A**) Design of xenograft experiments (*n* = 10/group); (**B**) the combination of ARN-3261 and carboplatin inhibits tumor growth in an OVCAR8 i.p. model. After treatment as indicated in (**A**) for three weeks, mice were weighed, and intraperitoneal nodules were excised and weighed. (**C**) Design of xenograft experiments (*n* = 10/groups); (**D**) The combination of ARN-3261 and primary chemotherapeutic drugs of carboplatin and paclitaxel inhibits tumor growth in an SKOv3 subcutaneous xenograft model. Mice were treated with single, double or triple agents for 6 weeks. Tumor was measured once a week until the tumor burden in control group reached maximum allowance. The graphs indicate the mean ± standard deviation. Statistical significance is indicated by * *p* < 0.05, ** *p* < 0.01 and *** *p* < 0.0001.

## Data Availability

No new data were created or analyzed in this study. Data sharing is not applicable to this article.

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
